# Clinical impact of hospital distance and center transfers on adherence and outcomes in familial adenomatous polyposis: A multicenter retrospective study in a defined region of Japan

Kyota Tatsuta[1], Mayu Sakata[1]*, Moriya Iwaizumi[2], Kazuya Okamoto[3], Shigeto Yoshii[4], Yutaro Asaba[5], Takashi Harada[6], Kiyotaka Kurachi[7], Mikihiro Shimizu[8], Hiroya Takeuchi[1]

**1** Department of Surgery, Hamamatsu University School of Medicine, Chuo-ku, Hamamatsu, Shizuoka, Japan, **2** Department of Laboratory Medicine, Hamamatsu University School of Medicine, Chuo-ku, Hamamatsu, Shizuoka, Japan, **3** Department of Surgery, Fujieda Municipal General Hospital, Fujieda, Shizuoka, Japan, **4** Department of Gastroenterology, Fujieda Municipal General Hospital, Fujieda, Shizuoka, Japan, **5** Department of Surgery, JA Shizuoka Kohseiren Enshu Hospital, Chuo-ku, Hamamatsu, Shizuoka, Japan, **6** Department of Surgery, Hamamatsu Medical Center, Chuo-ku, Hamamatsu, Shizuoka, Japan, **7** Department of Surgery, Hamamatsu Red Cross Hospital, Hamana-ku, Hamamatsu, Shizuoka, Japan, **8** Center for Clinical Research, Hamamatsu University School of Medicine, Chuo-ku, Hamamatsu, Shizuoka, Japan

* mayu-s@hama-med.ac.jp

**Editor:** Mari Kajiwara Saito, London School of Hygiene & Tropical Medicine Centre of Global Change and Health: London School of Hygiene & Tropical Medicine, UNITED KINGDOM OF GREAT BRITAIN AND NORTHERN IRELAND

## Abstract

This study aimed to evaluate how hospital distance and changes in surveillance hospitals influence adherence to surveillance, the cumulative risk of familial adenomatous polyposis-related tumors, and survival outcomes in patients with familial adenomatous polyposis. We conducted a multicenter retrospective study in a specific region of Japan and analyzed 79 patients with familial adenomatous polyposis who underwent total colectomy or proctocolectomy between 1987 and 2025 across 9 accredited hospitals. We examined the associations between straight-line distance to the hospital, changes in surveillance centers, and surveillance adherence, as well as the cumulative risk of familial adenomatous polyposis-related tumors and survival outcomes. The 10-year surveillance adherence rate was 86.5%. During follow-up, 31.6% of patients changed hospitals. Hospital distance did not differ significantly between those who maintained or dropped out of surveillance. However, patients residing ≥40 km from their hospital were significantly more likely to switch hospitals (61.5% vs. 13.2%, p<0.001). Importantly, changes in surveillance hospitals showed no significant association with the cumulative risk of familial adenomatous polyposis-related tumors or survival outcomes. Surveillance dropout occurred in 7.6% of patients. No specific clinical predictors of surveillance dropout were identified; the most common reason for surveillance dropout was patients' self-assessed low risk following negative screening results. This multicenter study found that hospital

**Data availability statement:** All relevant data are within the manuscript and its Supporting Information files.

**Funding:** The author(s) received no specific funding for this work.

distance or changes in surveillance hospitals did not significantly affect adherence to surveillance or clinical outcomes in patients with familial adenomatous polyposis.

## Introduction

Familial adenomatous polyposis (FAP) is an autosomal dominant hereditary disorder caused by a germline mutation in the adenomatous polyposis coli (*APC*) gene [1]. Without intervention, patients with FAP, who typically develop numerous colonic polyps by their second decade of life, face an almost certain progression to colorectal cancer by 40–50 years of age [2]. In addition to colorectal involvement, FAP is associated with extracolonic manifestations, including gastric fundic gland polyps and gastric or duodenal adenomas, all of which carry malignant potential [3]. Furthermore, desmoid tumors, a notable non-gastrointestinal complication, commonly arise in FAP, adding complexity to clinical management [4]. These FAP-related tumors significantly impact prognosis [5,6], highlighting the importance of comprehensive, lifelong surveillance strategies to improve long-term survival outcomes.

The effectiveness of surveillance programs depends not only on the quality of medical care but also on patients' adherence to regular follow-up appointments. Key factors such as commuting distance to the hospital and changes in the hospital conducting surveillance can significantly affect their ability to maintain consistent monitoring. A previous meta-analysis has underscored the negative influence of geographical barriers on healthcare access, particularly for patients with malignant tumors requiring regular follow-up care [7]. In patients with FAP, such logistical challenges can disrupt surveillance, potentially leading to delayed tumor detection, increased cancer risk, and poorer survival outcomes. Despite these implications, the impact of hospital distance and changes in the hospital conducting surveillance on adherence to monitoring protocols, oncological outcomes, and survival in patients with FAP remain insufficiently clear.

We hypothesized that longer commuting distances and frequent changes in surveillance hospitals would be associated with high dropout rates from recommended surveillance, high cumulative risk of FAP-related tumors, and low survival rates among patients with FAP. To test this hypothesis, we conducted a multicenter retrospective study analyzing data from specific regions in Japan. This study aimed to evaluate how hospital distance and changes in surveillance hospitals influence adherence to surveillance, the cumulative risk of FAP-related tumors, and survival outcomes in patients with FAP.

## Materials and methods

### Study design and patient population

This study utilized original data compiled from 9 institutions in Shizuoka Prefecture, Japan, all accredited by the Japan Surgical Society or the Japanese Society of Gastroenterology. Although no formal accreditation system has been established specifically for treating FAP in Japan, these accreditations ensure that each hospital

maintains a sufficient standard of surgical and gastrointestinal care to provide appropriate management for patients with FAP. Each institution collected and registered data on patients diagnosed with FAP between January 1987 and September 2024 for inclusion in the database. FAP was defined based on the following criteria: (i) the presence of 100 or more adenomatous polyps in the colon, regardless of family history; (ii) fewer than 100 adenomatous polyps in the colon, accompanied by a family history of FAP; or (iii) the presence of pathogenic germline variants in the *APC* gene [8]. This retrospective observational cohort study was approved by the institutional review boards (IRBs) of all participating institutions (IRB numbers: 24–190). Data for this retrospective study were accessed between October 1, 2024, and May 31, 2025; however, no additional cases registered during this period were included in the present analysis. During the process of data collection, the investigators had access to information that could identify individual participants.

## Surgery

Surgical treatment was generally indicated for patients with the dense type of FAP, characterized by ≥ 1001 colorectal polyps. For patients with the sparse (100–1000 colorectal polyps) or attenuated (≤ 99 colorectal polyps) type, surgical intervention was typically recommended but could be deferred in favor of non-surgical management based on patient preferences. Hand-sewn ileal pouch–anal anastomosis (IPAA) was the standard procedure. Segmental colectomy and total proctocolectomy with ileostomy were performed only in cases where curative resection was not feasible due to multiple distant metastases or extensive invasion of adjacent organs by colorectal cancer. These patients were excluded from this analysis, as they died within a few years due to advanced colorectal cancer. The detailed surgical procedures are described in previous studies [9,10]. Data extraction identified 79 cases who underwent total colectomy and proctocolectomy with confirmed surveillance information (Fig 1).

## Surveillance

Most patients underwent annual endoscopic surveillance, including esophagogastroduodenoscopy, colonoscopy, and pouchoscopy. In addition, computed tomography from the cervical to pelvic regions was performed every few years. The

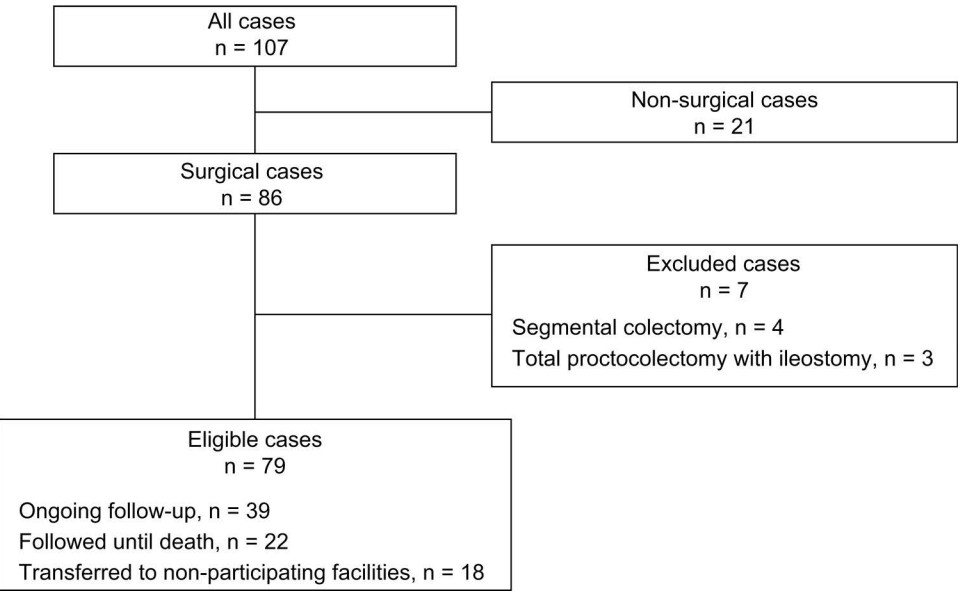

**Fig 1. Flow diagram of the study.**

frequency of these follow-ups varied depending on the era and type of surgical procedure performed. Surveillance pro-
tocols were determined individually by each institution. These protocols included patients who continue lifelong follow-up
at the same hospital and those who transitioned to a different hospital for surveillance owing to various reasons. The
analysis focused on comparing outcomes based on whether patients continued their surveillance at the same hospital or
changed institutions.

Surveillance continuation was defined as the period from the date of to completion of surveillance. Patients were moni-
tored until death, loss to follow-up, or end of the study (May 31, 2025). Patients who died during follow-up were censored
at the time of death. Those who transferred to facilities outside the participating centers were censored at the time of
transfer and were not followed thereafter. Those lost to follow-up or still under follow-up at the study's end were treated as
censored. The duration of surveillance continuation was measured in days until the point of censoring. Disease-specific
survival (DSS) was calculated from surgery until death, specifically due to FAP-related tumors. The cumulative risks of
colorectal cancer, gastric cancer, duodenal cancer, metachronous rectal cancer, and desmoid tumors were calculated up
to the respective dates of diagnosis. Metachronous colorectal cancer was defined using a 1-year cutoff according to the
JSCCR guideline [8]. The cumulative risks of colorectal, gastric, and duodenal cancers were estimated using age as the
time scale, whereas those of metachronous rectal cancer and desmoid tumors were estimated using years after surgery
as the time scale, to ensure consistency with their respective clinical onset timing. The cumulative risk of metachronous
rectal cancer was analyzed exclusively in patients with residual rectum, including those who underwent ileorectal anasto-
mosis and stapled IPAA.

## Distance to the hospital

For patients with a recorded residential address (n = 55), the straight-line distance to the hospital was measured using
Google Maps (https://www.google.co.jp/maps). When the exact residential address was unavailable, the distance was
measured from the municipal office, which was used as a proxy for the patient's location, to the hospital. For patients
who changed hospitals during follow-up, the distance to both the initial and final hospitals was calculated and used in the
corresponding analyses.

## Hospital changes in surveillance

We analyzed the association between hospital changes during surveillance and the distance to the hospital. To assess the
association between these changes and clinical outcomes, we divided the cases into two groups, (those with a change in
the surveillance hospital and those without), and conducted a comparative analysis.

## Analysis of factors that predict surveillance dropout

Surveillance dropout was defined as the absence of regular FAP-related medical consultations or examinations, including
routine visits or scheduled tests, for a period of 3 years or longer. This definition was applied only when such absence was
unequivocally confirmed. We analyzed the clinical factors associated with surveillance dropout and explored the under-
lying reasons for discontinued surveillance. Reasons for resuming FAP surveillance were reviewed from medical records
documented during routine outpatient visits at the time of surveillance resumption. To address potential limitations related
to the 3-year definition of surveillance dropout, sensitivity analysis was performed by restricting the cohort to patients who
were registered on or before May 31, 2022 (n = 77).

## Statistical analysis

Statistical analyses were conducted using JMP® 18 (SAS Institute Inc., Cary, NC, USA) and EZR version 1.6.3 (Saitama
Medical Center, Jichi Medical University, Saitama, Japan). Continuous variables were summarized as medians and

ranges and tested using the Mann–Whitney U test. Categorical data are presented as numbers and frequency and were analyzed using Fisher's exact test. The Kaplan–Meier method and log-rank testing were employed to calculate cumulative risk estimates and compare surveillance continuation, DSS, and cumulative risks for colorectal, gastric, and duodenal cancers, and desmoid tumors. In addition, the Gray test was performed to account for death prior to the event of interest as a competing risk. Univariable analyses of surveillance dropout were performed using logistic regression. A *p*-value of < 0.05 was considered statistically significant.

## Results

### Clinical characteristics

The characteristics of all study participants are summarized in Table 1. The median follow-up period was 12.3 years. The age distribution of patients at the time of FAP diagnosis and the final follow-up are illustrated in Fig 2. The Kaplan–Meier-estimated surveillance continuation rate was 86.5% (95% confidence interval [CI], 76.1–92.9) at 10 years and 64.4% (95% CI, 50.5–76.7) at 20 years (Fig 3).

### Distance to hospital

Surveillance continuation was categorized into three patterns: Pattern I, continuous surveillance at the same hospital (n = 54); Pattern II, initial surveillance at the diagnosing hospital, followed by transfer to another hospital, often closer to the patient's residence (n = 19); and Pattern III, surveillance at a hospital different from the diagnosing center from the outset (n = 6) (Fig 4A). Median hospital distances were 20.2 km (Pattern I), 73.3 km initially decreasing to 3.1 km after transfer (Pattern II), and 77.9 km decreasing to 7.4 km (Pattern III) (Fig 4B). Patterns II and III were analyzed as the "changed hospital group" in the following analyses.

### Association with distance to hospital

We examined how the distance to the hospital influenced the continuation of surveillance. In this database, six cases of surveillance dropouts were identified (Table 1). A comparison of hospital distances at the start of surveillance between individuals who continued surveillance and those who dropped out revealed no significant difference (Fig 5A). The analysis revealed that the distance to the hospital at the time of diagnosis was significantly longer in the changed hospital group compared with the same hospital group (Fig 5B), suggesting that hospital distance may influence decisions to change in hospitals. To explore this further, we examined the rate of surveillance hospital changes in relation to hospital distance. When the distance to the hospital exceeded 40 km, the rate of surveillance hospital changes was over 50% (Fig 5C). Comparison based on the 40 km threshold revealed a significantly higher likelihood of hospital change for patients 40 km or more from the hospital compared to those living closer (<40 km group: 9.09% vs. ≥40 km group: 62.86%; p < 0.001; Fig 5D).

### Association with changes in surveillance hospital

Next, we investigated how changes in the hospital where surveillance was conducted were associated with surveillance continuity, the cumulative risk of FAP-related tumors, and survival outcomes. Table 1 outlines the clinical characteristics, categorized by whether patients changed their surveillance hospital. A significant difference was observed between the groups regarding sex, whereas no significant differences were found in other factors. The intervals between upper gastrointestinal endoscopy, lower gastrointestinal endoscopy, and imaging tests were similar between the two groups, with no significant difference in dropout rates.

We compared the distribution and incidence of typical FAP-related tumors, including colorectal, gastric, and duodenal cancers, metachronous rectal cancer, and desmoid tumors, between the two groups (Table 1). The incidence of these tumors was similar regardless of whether the surveillance hospital was changed, although colorectal cancer tended to

**Table 1. Clinical characteristics.**

| | All cases<br>n = 79 | Same hospital group<br>n = 54 | Changed hospital group<br>n = 25 | P-value |
|---|---|---|---|---|
| Age at FAP diagnosis, years, median (range) | 32 (12–66) | 32 (12–66) | 26 (16–56) | 0.110 |
| Sex, cases (%) | | | | 0.007 |
| Male | 33 (41.8) | 17 (31.5) | 16 (64.0) | |
| Female | 46 (58.2) | 37 (68.5) | 9 (36.0) | |
| Family history, cases (%) | | | | 0.792 |
| Yes | 56 (70.9) | 39 (72.2) | 17 (68.0) | |
| No | 23 (29.1) | 15 (27.8) | 8 (32.0) | |
| Phenotype, cases (%) | | | | 0.270 |
| Profuse (polyps ≥1001) | 34 (43.0) | 25 (46.3) | 9 (36.0) | |
| Sparse (polyps 100–1000) | 42 (53.2) | 26 (48.1) | 16 (64.0) | |
| Attenuated (polyps ≤99) | 3 (3.8) | 3 (5.6) | 0 (0) | |
| Genetic test, cases (%) | | | | 0.809 |
| Yes | 34 (43.0) | 24 (44.4) | 10 (40.0) | |
| No | 45 (57.0) | 30 (55.6) | 15 (60.0) | |
| Colectomy, cases (%) | | | | 0.139 |
| Hand-sewn IPAA | 31 (39.2) | 23 (42.6) | 8 (32.0) | |
| Stapled IPAA | 23 (29.1) | 12 (22.2) | 11 (44.0) | |
| IRA | 25 (31.6) | 19 (35.2) | 6 (24.0) | |
| Median follow-up period, years (range) | 12.3 (0.21–45.4) | 11.8 (0.66–45.4) | 12.3 (0.21–24.1) | 0.483 |
| Interval for upper gastrointestinal endoscopy, years (range)* | 1.90 (0.13–10.25) | 1.75 (0.54–10.25) | 2.24 (0.13–8.71) | 0.565 |
| Interval for lower gastrointestinal endoscopy or pouchoscopy, years (range)* | 1.84 (0.13–20.99) | 1.77 (0.62–20.99) | 2.14 (0.13–7.45) | 0.689 |
| Intervals for imaging examinations, years (range)* | 1.59 (0.25–40.34) | 1.55 (0.25–40.34) | 2.04 (0.71–5.33) | 0.384 |
| Dropout, cases (%) | 6 (7.6) | 4 (7.4) | 2 (8.0) | 1.000 |
| FAP-related tumors, cases (%) | | | | |
| Colorectal cancer | 28 (35.4) | 23 (42.6) | 5 (20.0) | 0.078 |
| Metachronous rectal cancer | 10 (12.7) | 7 (13.0) | 3 (12.0) | 1.000 |
| Gastric cancer | 12 (15.2) | 9 (16.7) | 3 (12.0) | 0.487 |
| Duodenal cancer | 6 (7.6) | 4 (7.4) | 2 (8.0) | 1.000 |
| Thyroid cancer | 2 (2.5) | 2 (3.7) | 0 (0) | 1.000 |
| Pouch cancer | 2 (2.5) | 1 (1.9) | 1 (4.0) | 0.536 |
| Desmoid tumor | 16 (20.3) | 12 (22.2) | 4 (16.0) | 0.764 |
| Cause of death, cases (%) | | | | |
| FAP-related malignancies | 14 (17.7) | 12 (22.2) | 2 (8.0) | 0.205 |
| Colorectal cancer | 7 (8.9) | 7 (13.0) | 0 (0) | 0.091 |
| Metachronous rectal cancer | 5 (6.3) | 3 (5.6) | 2 (8.0) | 0.649 |
| The others | 2 (2.5) | 2 (3.7) | 0 (0) | 1.000 |

* Only patients who have obtained a medical examination history.

Upper gastrointestinal endoscopy: 'Same hospital' group, n = 37; 'switched hospital' group, n = 13.

Lower gastrointestinal endoscopy or pouchoscopy: 'Same hospital' group, n = 38; 'switched hospital' group, n = 13

Imaging examinations: 'Same hospital' group, n=37; 'switched hospital' group, n=10.

FAP, familial adenomatous polyposis; IPAA, ileal pouch–anal anastomosis; IRA, ileorectal anastomosis.

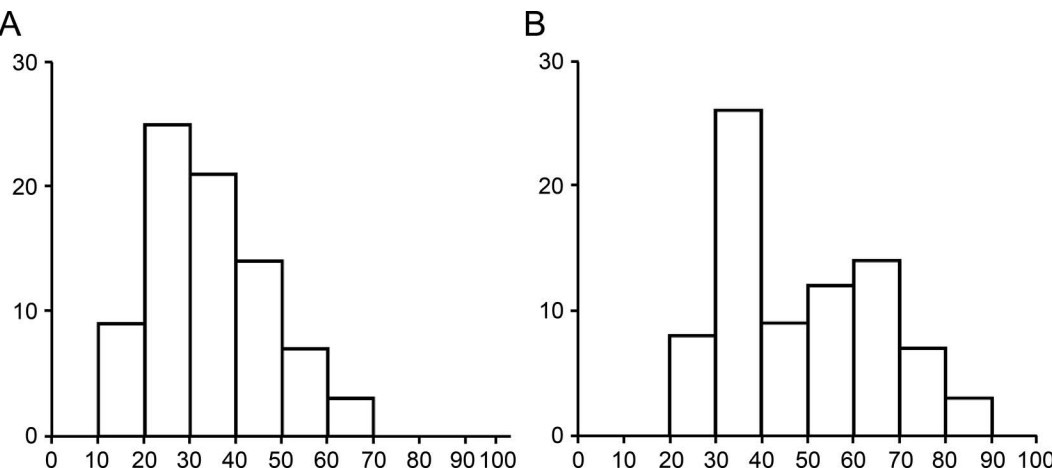

**Fig 2. Patient distribution. (A)** Age at the diagnosis of familial adenomatous polyposis; **(B)** Age at the final follow-up examination.

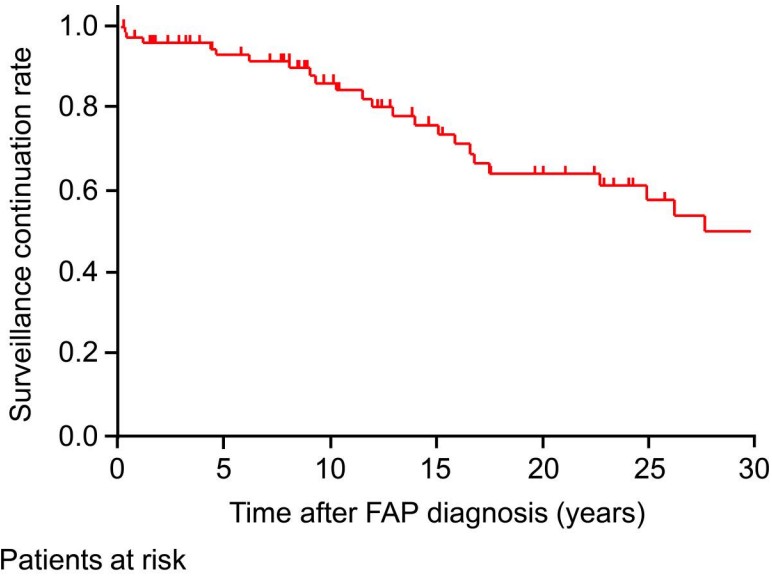

Patients at risk

| 79 | 64 | 48 | 33 | 25 | 17 | 14 |

**Fig 3. Surveillance continuation rate.**

occur more frequently. Cumulative risk estimates for these tumors, obtained using Kaplan–Meier analysis, showed no significant differences between the groups (Fig 6A–E). In addition, no significant differences were observed in FAP-related mortality or DSS between the two groups (Table 1 and Fig 6F). Similar results were obtained in the analysis using the Gray test, which accounted for death prior to the event of interest as a competing risk (S1 Fig).

**Clinical features that predict surveillance dropout**

To identify factors associated with surveillance discontinuation, we analyzed the clinical characteristics of patients who dropped out during the follow-up period. However, no specific factors were found to predict the clinical variables of

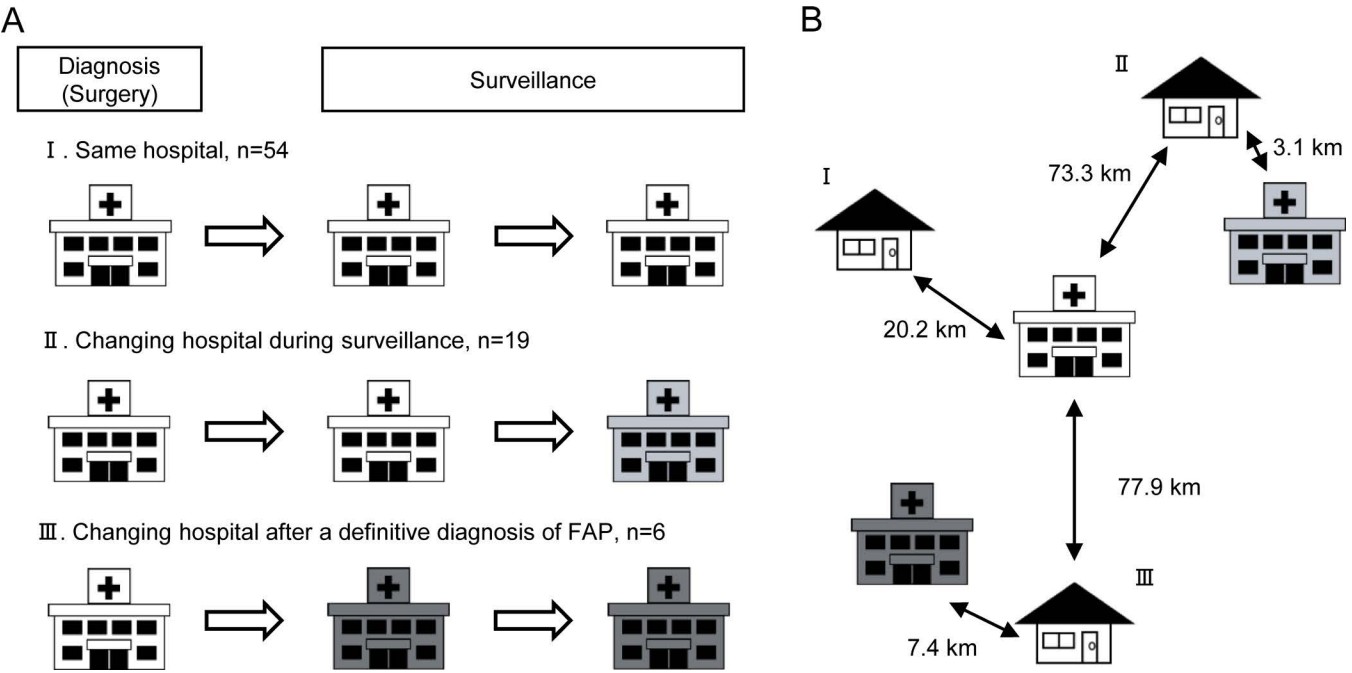

**Fig 4. Characteristics of distance to hospital in surveillance. (A)** Surveillance continuation status, 'Diagnosis' included cases where surgery was performed following the initial diagnosis. 'Surveillance' was defined as starting from the time of surgery; **(B)** Distance to hospital by each pattern.

surveillance dropout (Table 2). Upon further analysis of the dropout cases, we observed variability in the age at which dropout occurred, with a median suspension duration of 12.3 years (Table 3). As no common clinical characteristics were identified among patients who dropped out, we analyzed the reasons for their discontinuation. The most common reason was that patients, having experienced no issues in previous examinations, determined independently that further surveillance was unnecessary (Table 4). Sensitivity analysis yielded similar results, confirming robustness of these findings (S1 Table).

## Discussion

This multicenter retrospective study examined how logistical hospital-access factors, such as commuting distance and hospital changes, were associated with the continuation of surveillance in patients with FAP. We found that longer distances to the hospital at the time of diagnosis were associated with an increased likelihood of changing the hospital where surveillance occurred, particularly when the distance exceeded 40 km. However, we found that neither the distance to the hospital nor changes in the hospital for surveillance significantly influenced surveillance dropout, contrary to our initial hypothesis. Notably, these factors had no significant effect on the incidence of FAP-related tumors or survival. The primary reason for surveillance dropout was patients' self-perception of being low risk after previous negative results. To our knowledge, this is the first study to comprehensively explore the relationship between commuting distance and changes in the surveillance hospital among patients with FAP.

In general, geographical barriers are known to impair access to care and lead to delays in diagnosis or treatment [11]. In contrast, for patients with poor prognoses or rare cancers, treatment at specialized facilities can significantly improve outcomes, often mitigating the negative effects of travel distance [12,13]. These findings are based on studies from the United States and Europe, but nationwide analyses on hospital travel distances remain limited in Japan. In a study on patients with pediatric cancer, only 21.7% of patients required more than 1 hour of travel to reach a specialist hospital,

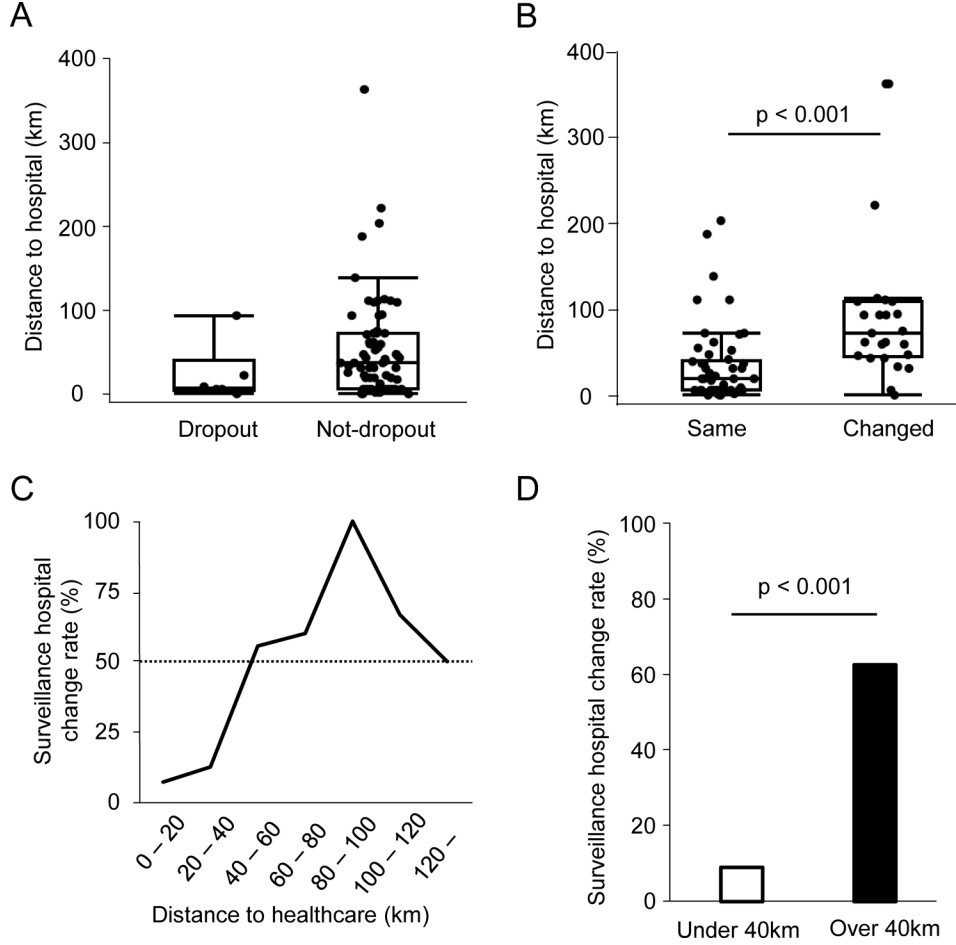

**Fig 5. Association between distance to hospital and surveillance. (A)** Distance to hospital at the start of surveillance compared between cases with and without dropout; **(B)** Distance to hospital at the time of diagnosis compared between the same and the changed hospital group; **(C)** The rate of surveillance hospital changes at the start of surveillance relative to the distance to the hospital; **(D)** Surveillance hospital change rates compared by a 40 km threshold.

with no significant difference in healthcare access between urban and rural areas [14]. However, the impact of proximity and travel time to hospitals on disease outcomes remains underexplored. This study is novel in its detailed investigation of the relationship between disease outcomes, distance to healthcare facilities, changes in hospitals, and surveillance continuity. In contrast, a multicenter study in Japan involving patients with FAP who underwent surgical treatment found a correlation between hospital volume, as measured by the number of surgeries performed, and overall survival [15]. A nationwide study is needed to further assess the equity of FAP treatment across Japan, with a particular focus on non-surgical cases, regional differences between urban and rural areas, and disparities in care between FAP-specialized and non-specialized hospitals.

While the proportion of patients who discontinued surveillance was relatively small (7.6%), it warranted further examination of the underlying reasons. Previous studies have identified factors such as older age, lack of recent surgery, and no history of malignancy as contributing to surveillance dropout [16]. However, this study found no significant association between these factors and surveillance dropout. Instead, the most common reason for surveillance dropout was patients' self-perception of low risk after receiving negative results in previous screenings. This result is consistent with a previous

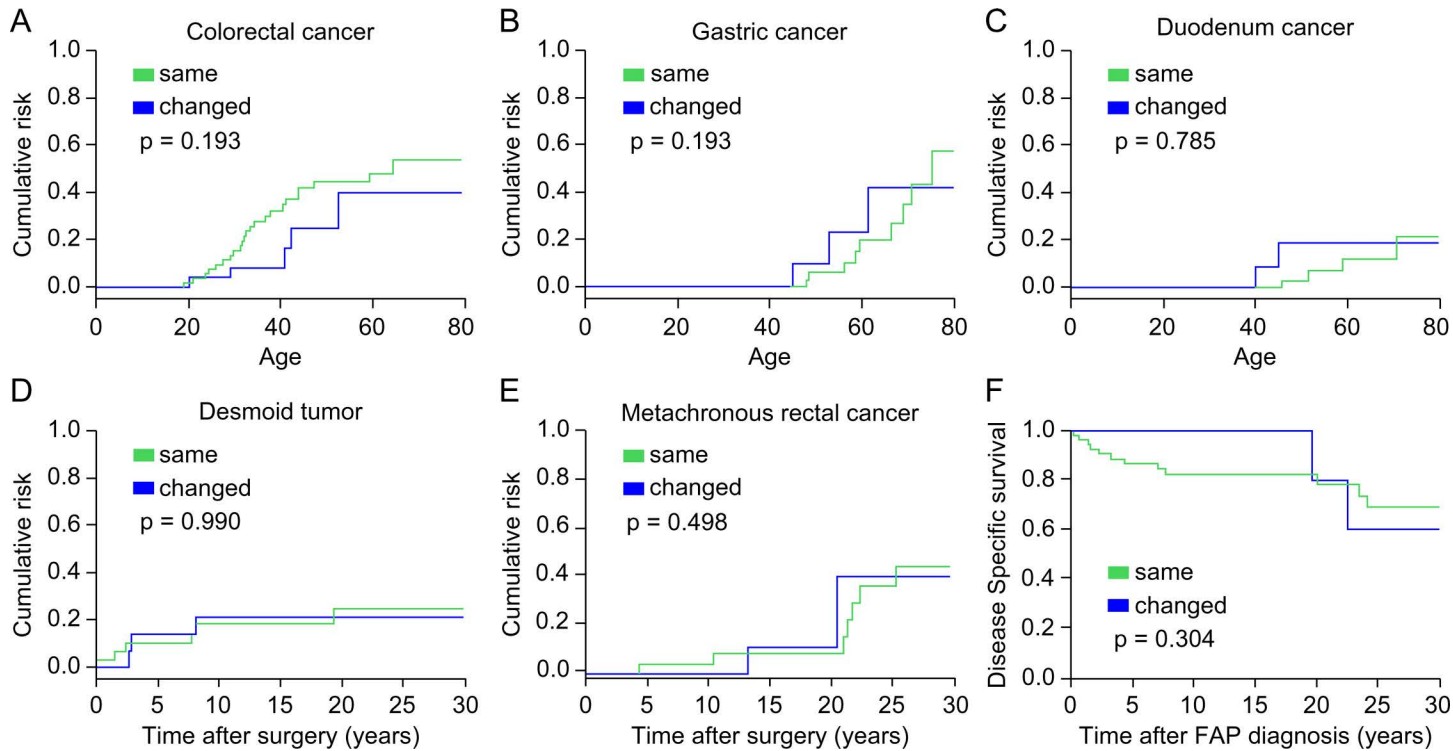

**Fig 6. Comparison of cumulative risk of FAP-related tumors and survival. (A–E)** Cumulative risk of FAP-related tumors, including (A) colorectal cancer, (B) gastric cancer, (C) duodenal cancer, (D) desmoid tumor, and (E) metachronous rectal cancer; **(F)** Disease-specific survival. FAP, familial adenomatous polyposis.

**Table 2. Univariate analysis of surveillance dropout.**

| | OR | 95% CI | *P*-value |
|---|---|---|---|
| Age, ≥ 32 years | 0.460 | 0.061–2.511 | 0.374 |
| Sex, men | 0.256 | 0.013–1.693 | 0.256 |
| Family history, yes | 0.377 | 0.065–2.184 | 0.262 |
| Genetic test, yes | 1.355 | 0.237–7.753 | 0.721 |
| Initial colorectal cancer, yes | 1.106 | 0.202–8.366 | 0.910 |
| Surveillance hospital change. yes | 1.087 | 0.143–5.993 | 0.927 |
| Distant to hospital, ≥ 40 km | 0.229 | 0.012–1.515 | 0.136 |

OR, odds ratio; CI, confidence interval.

report indicating that patients who do not adhere to surveillance often hold a more optimistic view of FAP [16,17]. Patients with FAP generally express concern regarding the hereditary and lifelong nature of the condition [18]. These contrasting perspectives highlight the need for personalized support systems to enhance surveillance adherence and improve patients' quality of life. To address these challenges, we are currently working to establish a regional hospital collaboration network that facilitates seamless information sharing among healthcare providers. As part of this effort, it is essential to define clear goals that promote timely specialist access and implement patient-centered strategies—such as tailored educational interventions—to bridge the perceptual gap between providers and patients and to foster long-term engagement with surveillance programs.

**Table 3. Details of patients who dropped out of surveillance.**

| | Age at FAP diagnosis, years | Sex | Surgery | Age at the final follow-up examination before drop-out, years | Duration of the surveillance dropout, years | Reason for return to surveillance | Diagnosis at initial examination after return to surveillance |
|---|---|---|---|---|---|---|---|
| Case 1 | 19 | Male | Stapled IPAA | 21 | 11 | Anal pain | Metachronous rectal cancer |
| Case 2 | 19 | Female | IRA | 35 | 5 | Occult blood in stool | Metachronous rectal cancer |
| Case 3 | 20 | Female | Hand-sewn IPAA | 29 | 28 | Anal pain | Pouch cancer |
| Case 4 | 31 | Female | Stapled IPAA | 44 | 18 | Family genetic counseling visit | No apparent |
| Case 5 | 40 | Female | Hand-sewn IPAA | 46 | 5 | Occult blood in stool | No apparent |
| Case 6 | 59 | Female | Hand-sewn IPAA | 74 | 7 | Occult blood in stool | Metachronous rectal cancer |

FAP, familial adenomatous polyposis; IPAA, ileal pouch–anal anastomosis; IRA, ileorectal anastomosis

**Table 4. Reasons for surveillance dropout.**

| Reason | n = 6 |
|---|---|
| Having had no issues in previous examinations, they considered further examination unnecessary, cases (%) | 3 (50.0) |
| Do not wish to have an examination due to advanced age, cases (%) | 1 (16.7) |
| No money to have an examination, cases (%) | 1 (16.7) |
| Refused to undergo examination, cases (%) | 1 (16.7) |

This study has some limitations. First, its retrospective design and small sample size may affect the reliability of the conclusions. Additionally, the participants were recruited from patients who were receiving care at medical facilities within a single geographic region, which limits the generalizability of the findings. However, the cumulative risk of FAP-related tumors and prognosis in this study are consistent with the results of a national study in Japan [19–24]. Second, the distance to the hospital was measured as the straight-line distance between the hospital and the patient's residence or the municipal office. Although the study period was extensive, making it challenging to account for changes in transportation infrastructure over time, assessing the actual travel time to the hospital would have been more appropriate. Third, the outcomes of patients who were transferred to non-participating facilities could not be observed, which may have resulted in incomplete follow-up data for a subset of cases.

## Conclusions

In conclusion, this study revealed that neither the distance to the hospital nor the changes in the hospital showed a significant association with surveillance dropout or FAP-related outcomes.

## Supporting information

**S1 Fig. Comparison of cumulative risk of FAP-related tumors and survival using the Gray test.**
(DOCX)

**S1 Table. Univariate analysis of surveillance dropout in the sensitivity analysis.**
(DOCX)

**S1 File. STROBE_Checklist.**
(DOCX)

**S2 File.**
(XLSX)

## Acknowledgments

The authors would like to acknowledge all the patients and their families. In addition to the investigators listed as authors, we also acknowledge the following investigators who contributed to the study: Takahiro Uotani, Department of Gastroenterology, Japanese Red Cross Shizuoka Hospital; Yasuyuki Kobayashi, Department of Surgery, Seirei Hamamatsu General Hospital; Takahiro Koizumi, Department of Surgery, Omaezaki Municipal Hospital; Atsuko Fukazawa, Department of Surgery, Iwata City Hospital; Takanori Yamada, Department of Gastroenterology, Iwata City Hospital; Koichi Nakamura, Department of Surgery, Kikugawa General Hospital. We used ChatGPT (OpenAI) to refine the grammar and polish the language of the manuscript.

## Author contributions

**Conceptualization:** Mayu Sakata.

**Data curation:** Kyota Tatsuta.

**Formal analysis:** Kyota Tatsuta, Mikihiro Shimizu.

**Investigation:** Kyota Tatsuta, Kazuya Okamoto, Shigeto Yoshii, Yutaro Asaba, Takashi Harada, Kiyotaka Kurachi.

**Methodology:** Kyota Tatsuta, Mayu Sakata.

**Project administration:** Mayu Sakata, Hiroya Takeuchi.

**Resources:** Moriya Iwaizumi, Kazuya Okamoto, Shigeto Yoshii, Yutaro Asaba, Takashi Harada, Kiyotaka Kurachi.

**Software:** Kyota Tatsuta.

**Supervision:** Mayu Sakata, Hiroya Takeuchi.

**Validation:** Kyota Tatsuta, Moriya Iwaizumi.

**Writing – original draft:** Kyota Tatsuta.

**Writing – review & editing:** Mayu Sakata, Kiyotaka Kurachi, Hiroya Takeuchi.

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
