## [Decision Letter · Decision Letter 0]

15 Oct 2025

Dear Dr. Sakata,

Thank you for submitting your manuscript to PLOS ONE. After careful consideration, we feel that it has merit but does not fully meet PLOS ONE’s publication criteria as it currently stands. Therefore, we invite you to submit a revised version of the manuscript that addresses the points raised during the review process.

We look forward to receiving your revised manuscript.

Kind regards,

Mari Kajiwara Saito, M.D., Ph.D.

Academic Editor

PLOS ONE

2. We note that your Data Availability Statement is currently as follows: [The data are all contained within the manuscript and/or Supporting Information files.]

Additional Editor Comments:

Major comments: Have authors used AIs for drafting this manuscript? If so, it would be nice if the authors could include the statement on the use of generative AIs (which AI, in which step) in Acknowledgement section.

I assume the following is likely to be AI-generated text, because these phrases seem to have lost your tone and voice: L52-56, L60-64, p16 the last two lines-p17 the first two lines

All authors of this manuscript have responsibility for its originality. Generative AI cannot be an author, and verbatim (blanketly just copying and pasting the translated phrases) from generative AI is ethically contingent, may not be acceptable, and authors could be in danger of plagiarism without any intention. To maintain academic competency and research integrity, it is advisable that authors use generative AIs appropriately for this and further research.

Cheng A, Calhoun A, Reedy G. Artificial intelligence-assisted academic writing: recommendations for ethical use. Adv Simul (Lond). 2025;10(1):22.

Minor comments: the word “self-determined” has a different meaning. Please consider changing the word.

Reviewers' comments:

Reviewer's Responses to Questions

**Comments to the Author**

1. Is the manuscript technically sound, and do the data support the conclusions?

Reviewer #1: Partly

2. Has the statistical analysis been performed appropriately and rigorously?

Reviewer #1: No

3. Have the authors made all data underlying the findings in their manuscript fully available?

Reviewer #1: Yes

4. Is the manuscript presented in an intelligible fashion and written in standard English?

Reviewer #1: Yes

Reviewer #1: Thank you for the opportunity to review this manuscript. I found the topic to be important and the effort to collect multicenter data commendable. However, there are several concerns regarding methodology, definitions, and clarity of reporting. Due to the length of my review, I have provided detailed comments in the attached document for the authors’ and editors’ consideration.

**Do you want your identity to be public for this peer review?** For information about this choice, including consent withdrawal, please see our Privacy Policy

Reviewer #1: No

---

## [Author Response · Author response to Decision Letter 1]

25 Nov 2025

Response to Reviewer file uploaded in Attach files section.

---

## [Decision Letter · Decision Letter 1]

8 Dec 2025

Clinical impact of hospital distance and center transfers on adherence and outcomes in familial adenomatous polyposis: A multicenter retrospective study in a defined region of Japan

PONE-D-25-45810R1

Dear Dr. Sakata,

We’re pleased to inform you that your manuscript has been judged scientifically suitable for publication and will be formally accepted for publication once it meets all outstanding technical requirements.

Kind regards,

Mari Kajiwara Saito, M.D., Ph.D.

Academic Editor

PLOS One

Additional Editor Comments (optional):

Congratulations on the acceptance of your study. I have a few minor comments on the revised manuscript.

1. L178: Univariate analyses of surveillance dropout were performed using logistic regression. & Table 2, S1 Table

I am uncertain whether the authors used the term “univariate” in its conventional statistical meaning. Please clarify whether the term was intended to refer to a single outcome (univariate) or whether “univariable” (indicating a single exposure) would be more appropriate.The authors state that logistic regression was used, but in the Table 2 and S1 Table, hazard ratios are reported. Please clarify which statistical model was actually used.

2. The definition of metachronous CRC varies across countries and studies. I reckon the authors intended to define metachronous CRC using a 1-year cutoff, as specified by the Japanese Society for Cancer of the Colon and Rectum. Please specify how the metachronous CRC was defined in this study and provide an appropriate citation.

Reviewers' comments:

Reviewer's Responses to Questions

**Comments to the Author**

Reviewer #1: (No Response)

2. Is the manuscript technically sound, and do the data support the conclusions?

Reviewer #1: (No Response)

3. Has the statistical analysis been performed appropriately and rigorously?

Reviewer #1: (No Response)

4. Have the authors made all data underlying the findings in their manuscript fully available?

Reviewer #1: (No Response)

5. Is the manuscript presented in an intelligible fashion and written in standard English?

Reviewer #1: (No Response)

Reviewer #1: (No Response)

**Do you want your identity to be public for this peer review?** For information about this choice, including consent withdrawal, please see our Privacy Policy

Reviewer #1: No

---

## [Editor Report · Acceptance letter]

PONE-D-25-45810R1

PLOS One

Dear Dr. Sakata,

I'm pleased to inform you that your manuscript has been deemed suitable for publication in PLOS One. Congratulations! Your manuscript is now being handed over to our production team.

Kind regards,

on behalf of

Dr. Mari Kajiwara Saito

Academic Editor

PLOS One